

# Habituation but not classical conditioning of the disturbance hiss of the hissing cockroach (*Gromphadorhina portentosa*)

Christopher A. Varnon

University of North Texas, Denton, TX, United States

## ABSTRACT

This article explores learned changes in the disturbance hiss of the hissing cockroach, *Gromphadorhina portentosa*. Compared to extensive research on learning in other cockroaches, studies with this species are rare. Of the natural behaviors of *G. portentosa*, the disturbance hiss is also seldom investigated. Two experiments were conducted to address these deficits. The first experiment investigated habituation to repeated tactical stimulus delivered near the cerci. The effect of sex and heat were also assessed in a group design. This experiment found typical habituation trends, with males showing higher rates of hissing, and heated cockroaches showing marginally higher rates of hissing. Similar, but less pronounced results were seen with probability of movement. The second experiment explored classical conditioning by presenting an olfactory stimulus prior to, and along with, tactile stimulation. After conditioning, the olfactory stimulus and a second novel olfactory stimulus were presented on opposite ends of the apparatus to determine if there was conditioned preference. No evidence of conditioned response was observed in this experiment. Hissing and movement were observed during and after tactile stimulation, but responses were not observed before trials or during olfactory stimulus presentations. No preference between novel and conditioned odor was observed in the preference test. These findings confirm habituation in *G. portentosa* but highlight challenges in eliciting conditioned responses, emphasizing the need for further research to enhance understanding of insect learning and behavior.

## INTRODUCTION

This article investigates if the hissing cockroach, *Gromphadorhina portentosa*, can alter its disturbance hiss through learning. *G. portentosa* is the most well-known of several Madagascar hissing cockroach species in the tribe Gromphadorhini. These large, wingless cockroaches produce several hissing sounds by forcing air from their respiratory spiracles (*Nelson, 1979*; *Nelson & Fraser, 1980*). While several species of cockroaches produce sounds through stridulation, often when disturbed (*Guthrie, 1966*; *Roth & Hartman, 1967*; *Schal, Fraser & Bell, 1982*), this mechanism of sound production is unique to the tribe Gromphadorhini. Adult males can produce agonistic and courtship hisses, and older nymphs and adults of both sexes can produce a disturbance hiss in response to either visual

Corresponding author
Christopher A. Varnon,
christopher.varnon@unt.edu

or tactical stimuli. A weak stimulus may produce a single hiss, while a strong stimulus may cause a long series of hisses as the animal flees (*Nelson & Fraser, 1980*).

There is growing interest in the behavior and learning abilities of *G. portentosa*, both as subjects of experimental research and as practical species for hands-on teaching opportunities. This can be seen through student thesis projects (*e.g.*, *Albaitis, 2022*; *Gunnarsson, 2013*; *Harri-Dennis, 2016*) and numerous presentations at behavior-focused conferences such as the Association of Behavior Analysis International. The interest in *G. portentosa* as practical model for behavior mirrors interest in this species as a model for anatomy (*Heyborne, Fast & Goodding, 2012*), immunology (*Chua et al., 2017*), and general science teaching (*Fisher & Lorenz-Reaves, 2018*; *Wagler & Wagler, 2011*, *2019*, *2021*).

Unfortunately, many aspects of this species' natural ecology and behavior are not well understood. Historically, much of the ecological research on *G. portentosa* has been limited to descriptions of geographic distributions (*Schal, Gautier & Bell, 1984*). While the agonistic and courtship hisses have been thoroughly investigated in the laboratory (*e.g.*, *Clark & Moore, 1995a*, *1995b*, *1995c*; *Fraser & Nelson, 1984a*, *1984b*), the disturbance hiss, although well-known, is seldom studied. One possibility is that it serves as a deimatic response that startles a potential predator. While these often involve visual displays, auditory deimatic responses are not unusual in insects (*Drinkwater et al., 2022*), as in whistling of walnut sphinx caterpillars (*Bura et al., 2011*). Alternatively, the disturbance hiss may function as a pursuit deterrence signal, warning a predator that it has been detected (*Hasson, 1991*). For instance, white-tailed deer raise their tails to expose a bright white patch before fleeing (*Bildstein, 1983*), and bumblebees lift one or multiple legs when disturbed, prior to stinging (*Varnon et al., 2021*). The disturbance hiss may also act as a warning for other cockroaches, as research shows they do respond to the sounds of other types of hisses (*Nelson & Fraser, 1980*). Finally, it is possible the disturbance hiss is unrelated to predation attempts. One study showed that contact with a potential predator caused immobility rather than hissing, and that the disturbance hiss occurred more frequently in social contexts (*Shotton, 2014*).

Published work on learning in this species is also exceedingly rare. There are only two published studies. *Davis & Heslop (2004)* reported habituation of the disturbance hiss in response to handling in ten out of twelve cockroaches (another eight never hissed), with four showing a recovery of hissing in the presence of a novel handler. Additionally, *Dixon et al. (2016)* demonstrated individual food preferences in seven cockroaches using a reinforcement procedure. While these studies are useful, the small number of publications is in great contrast to other cockroach species. Topics include studies of habituation (*Varnon & Adams, 2021*; *Zilber-Gachelin & Chartier, 1973a*, *1973b*), classical conditioning (*Lent & Kwon, 2004*; *Varnon, Barrera & Wilkes, 2022*; *Watanabe et al., 2003*; *Watanabe & Mizunami, 2007*), operant conditioning (*Arican et al., 2020*; *Garren, Sexauer & Page, 2013*), spatial learning (*Brown & Strausfeld, 2009*), learned helplessness (*Brown & Stroup, 1988*), memory formation (*Hosono, Matsumoto & Mizunami, 2006*), and neuroscience (*Matsumoto et al., 2013*; *Sato, Matsumoto & Mizunami, 2023*). While most work involves *Periplaneta americana* or *Blattella germanica*, two well-known pests (*Schal, Gautier & Bell, 1984*), several studies involve non-pest species that may be more practical for many

laboratories, such as *Blaberus cranifer* (*Zilber-Gachelin & Chartier, 1973a*, *1973b*), *Blaberus discoidalis* (*Harley, English & Ritzmann, 2009*), *Eublaberus posticus* (*Varnon & Adams, 2021*; *Varnon, Barrera & Wilkes, 2022*), *Nauphoeta cinerea* (*Kou et al., 2019*; *Longo, 1964*) and *Rhyparobia maderae* (*Garren, Sexauer & Page, 2013*). Clearly, there is potential to study cockroach behavior and learning. What is missing is research with *G. portentosa*, both in terms of the disturbance hiss and in terms of learning. This article describes two experiments designed to fill some of these gaps.

The first experiment explored habituation of the disturbance hiss. Habituation is highly conserved and occurs across a range of animals, including rodents (*Davis, 1974*), songbirds (*Vincze et al., 2016*), snakes (*Place & Abramson, 2008*), bees (*Varnon et al., 2021*), and single-celled organisms (*Rajan et al., 2023*). While often studied in cockroaches, *Davis & Heslop (2004)* is the only investigation of habituation in *G. portentosa*. The current experiment confirms and extends that work by presenting cockroaches with repeated tactile stimuli to elicit habituation of the disturbance hiss and related behaviors, while also examining the roles of temperature and sex.

The effect of temperature was investigated as it has a pronounced influence on the physiology and behavior of insects and other ectotherms. Although *G. Portentosa* is a tropical species often studied in heated conditions around 27 °C (*e.g.*, *Dixon et al., 2016*; *Nelson & Fraser, 1980*). It is tolerant of typical indoor temperatures and has been studied at lower temperatures around 22 °C (*e.g.*, *Davis & Heslop, 2004*). However, as temperature approaches 15 °C, oxygen consumption and heart rate decrease (*Herreid, Full & Prawel, 1981*; *Streicher, Cox & Birchard, 2012*). Such physiological changes may directly alter behavior, while further changes in behavior may arise as adaptive responses to cooler conditions (*Abram et al., 2017*). While the effect of temperature on learning in cockroaches is not widely explored, one study with *Blatella germanica* found that cooler temperatures (~4.5 °C) inhibited learning and memory (*Hunter, 1932*). Generally, it appears that warmer individuals have a higher rate of behavior, and manipulation of temperature may provide a way to identify ideal conditions for learning. The effect of sex was investigated as there are known sex differences in hissing behavior. Only males produce agonistic and courtship hisses, and some sex differences in disturbance hiss in *G. portentosa* and related hissing cockroaches have been observed (*Hunsinger et al., 2018*; *Shotton, 2014*). Pilot work and classroom activities also suggested males produce the disturbance hiss notably more. Investigation of sex would allow confirmation of this difference.

The second experiment investigated classical conditioning. Like habituation, classical conditioning is a widespread phenomenon that has been documented across taxa, including rats (*Rescorla, 1968*), pigeons (*Hittesdorf & Richards, 1978*), fish (*Tennant & Bitterman, 1975*), and bees (*Abramson et al., 2015*). Unlike habituation, there are many species-specific trends in classical conditioning, especially when surrounding phenomena like autoshaping (*Jenkins & Moore, 1973*; *Palm & Powell, 1985*; *Timberlake & Grant, 1975*; *Wasserman, 1973*) and conditioned taste aversion (*Braveman, 1975*; *Garcia & Koelling, 1966*; *Ratcliffe, Fenton & Galef, 2003*; *Varnon et al., 2018*). Although there are many demonstrations of classical conditioning in cockroaches, including olfactory conditioning studies (*e.g.*, *Varnon, Barrera & Wilkes, 2022*; *Watanabe et al., 2003*), there is no

documentation of classical conditioning in *G. portentosa*, nor is there research on olfactory aversive learning in cockroaches similar to those found in other insects such as ants (*Desmedt et al., 2017*), bees (*Carcaud et al., 2009*; *Tedjakumala & Giurfa, 2013*; *Vergoz et al., 2007*) and fruit flies (*Busto, Cervantes-Sandoval & Davis, 2010*). The absence of such findings, especially in a well-known species, raises the questions of whether such demonstrations of learning are possible and what they might look like. This topic was investigated in the second experiment, where an olfactory conditioned stimulus (CS) was presented alongside a tactile unconditioned stimulus (US) to determine if a disturbance hiss would emerge as a conditioned response (CR). As with the first experiment, sex differences were also investigated.

# MATERIALS AND METHODS

## Subjects

Adult hissing cockroaches (*Gromphadorhina portentosa*) were used as subjects for this experiment ($n = 160$). Prior to collection, cockroaches lived in two large breeding colonies. Founding members were obtained from Fluker's Farms (Port Allen, LA) and Rainbow Mealworms (Compton, CA). Subjects were at least second-generation descendants of founding members and were collected at least 1 year after the colonies were established. Each colony was maintained at 23 °C and 57% relative humidity in ventilated plastic bins (52 cm × 36 cm × 36 cm) with a layer of ReptiBark substrate (Zoo Med Laboratories, San Luis Obispo, CA, USA). A thin layer of petroleum jelly was applied to the top of the bins to prevent young cockroaches from escaping. Colonies were fed dry dog food (Purina One, Nestlé Purina PetCare, St. Louis, MO, USA), apples, and water *ad libitum*. Wooden and cardboard shelters were provided for the cockroaches to hide in and climb on. Colonies were maintained on day:night cycle, with two red 3-watt light bulbs (Feit Electric, Pico Rivera, CA, USA) providing illumination for the experimenters. Red lights do not disrupt the behavior of cockroaches as they cannot easily see red wavelengths (*Briscoe & Chittka, 2001*). Interaction with animals in the breeding colonies was limited to husbandry and subject collection. Lotions, perfumes, and other fragrances were prohibited in the laboratory to prevent cockroaches being affected by odors during husbandry, collection, or experimental sessions. Subjects were collected by gently lifting the animals by the thorax and upper abdomen. Only adults with intact antennae and legs were collected. After participating in the experiment, subjects were recollected and retired to a colony reserved for experimentally experienced subjects. All collection, handling, and interaction was conducted by a single experimenter.

## Experiment 1: habituation

Subjects ($n = 80$) were randomly selected from a breeding colony in 2015 one day before their participation in the experiment. Subjects were sexed and weighed during collection. Females weighed an average of 8.23 g (standard deviation (SD) = 1.35, range = 5.00–11.00), while males weighed an average of 6.88 g (SD = 1.90, range = 4.00–11.00). Subjects were then placed in individual apparatuses. Each apparatus was a ventilated plastic bin (14 cm × 12 cm × 7.5 cm) with a thin layer of ReptiBark substrate. The presence of hissing

when the subject was collected and moved to the apparatus was recorded. Subjects were allowed a one-day acclimation period in the apparatus.

Subjects were divided into groups based on temperature manipulation and sex. Heated subjects received the same procedure as unheated subjects, except that their apparatus was placed on an Intellitemp Reptile Heat Mat (Big Apple Pet Supply, Boca Raton, FL, USA). The surface temperature of each subject was recorded after the one-day acclimation period with a CH-1022-NCIR infrared thermometer (Engineering Products, Loganville, GA, USA) held 2 cm away from the mid abdomen. The average temperature, including initial and final temperatures, was 20.86 °C ($SD$ = 1.22, range = 18.86–26.59) for unheated subjects and 32.84 °C ($SD$ = 2.15, range = 27.22–36.53) for heated subjects. Unheated and heated groups were further divided by sex, leading to a 2 × 2 design with four groups of 20 subjects each.

The experiment began after the one-day acclimation period and consisted of 14 trials for each subject, starting 1 h after the end of the dark cycle. Five minutes prior to the first trial, the apparatus lid was removed to allow for observation and interaction. During each trial, the experimenter presented a tactile stimulus consisting of tapping the terminal abdominal segment, as close to the cerci as possible, with a small, flexible metal wire. This stimulus was selected as pilot work indicated it produced hissing reliably without requiring excessive force. After presenting the tactile stimulus, the experimenter and one observer independently recorded behavior for 10 s, then the subject was left undisturbed for a 6-min intertrial interval. After all trials were complete, subjects were recollected and placed in the colony for experimentally experienced cockroaches. The presence of hissing during recollection was also recorded. A video demonstrating the overall procedure can be seen in the Supplemental Material.

Two responses were recorded for each trial: (1) the number of hisses, and (2) the presence of movement. Hissing was defined as any audible vocalization that occurred within 10 s of the stimulus presentation. Duration and intensity of hissing varied but were not recorded. Hisses were initially counted by ear and recorded on paper during the experiment, then later verified by audio recording. Movement was defined as any time the subject took a step forward within 10 s of the stimulus presentation. Subjects were generally stationary before each trial, but movement was not recorded if subjects were moving at the start of the trial.

## Experiment 2: classical conditioning

Subjects ($n$ = 80) were randomly selected from a breeding colony in 2020 on the day of their participation in the experiment. Subjects were sexed and weighed during collection. Females weighed an average of 6.29 g (SD = 1.81, range = 2.95–10.70), while males weighed an average of 5.03 g (SD = 1.21, range = 2.79–7.89). Subjects were then placed in individual clear plastic, open top runway apparatuses (23.5 cm × 8.1cm × 7.5 cm). The upper rim of the runway was lined with petroleum jelly to prevent escape. Between conditioning trials, the runway was placed on an Intellitemp heat mat. During conditioning trials, the runway was placed on a paper grid covering a second heat mat between two fume extractors (Xytronic-USA, Shingle Springs, CA, USA). The fume extractors were used to remove

lingering odors, a standard precaution for insect olfactory conditioning procedures conducted in an open apparatus (*e.g.*, *Abramson et al., 2015*; *Desmedt et al., 2017*; *Matsumoto et al., 2012*; *Vergoz et al., 2007*). The heat mats maintained an average apparatus temperature of 31.48 °C ($SD$ = 2.28, range = 27.06–34.44). The paper grid underneath the apparatus contained marks that divided the runway into five 4.7 cm cells used to record movement. A camera positioned above the runway recorded each trial for later scoring by two independent observers. After finishing an experimental session, runways were cleaned with alcohol wipes and left to dry.

Each subject participated in eight conditioning trials following a 1-h acclimation period. During each trial, the runway was moved into the conditioning arena, and the subject was observed for a 30-s pre-trial interval. Next, the CS was delivered. The experimenter held a two-by-two cm filter paper soaked in 0.02 mL of orange or peppermint extract (McCormick, Hunt Valley, MA, US) near the subject's antennae. These odors were selected as they have been successfully used in other cockroach conditioning experiments (*Varnon, Barrera & Wilkes, 2022*; *Watanabe et al., 2003*), and hissing cockroaches appeared responsive to these odors (*i.e.*, changes in antennae movement and orientation) during student projects and classroom demonstrations. The odor used as a CS was counterbalanced across subjects. The CS odor was presented for either three or 6 s, depending on group assignment. The subject was observed for the CS interval, then the US was presented. The US delivery method followed the procedure of attaching the CS odor to the US delivery device described in *Hosono, Matsumoto & Mizunami (2006)*. Here, instead of attaching the filter paper to a syringe for food delivery, the filter paper was attached to a metal ring worn on the experimenter's hand, which delivered the tactile US. The US consisted of gently lifting the subject one cm by the thorax and upper abdomen for 3 s. If the subject dislodged itself, it was quickly lifted again and held until the 3 s US duration ended. During this time, the CS odor remained present. The temporal parameters of CS and US presentations in this experiment were used as they fall within the range of procedures common to olfactory conditioning research in cockroaches (*e.g.*, *Varnon, Barrera & Wilkes, 2022*; *Watanabe et al., 2003*), bees (*e.g.*, *Abramson et al., 2015*; *Frost, Shutler & Hillier, 2012*), and other insects (*e.g.*, *Busto, Cervantes-Sandoval & Davis, 2010*; *Desmedt et al., 2017*). After the 3-s US interval, the subject was observed for a 30-s post-trial interval. The runway was then returned to its position on the first heat mat for a 30-min intertrial interval. A video demonstrating the overall procedure can be seen in the Supplemental Material.

During the pre-trial, CS, US, and post-trial intervals, the number of hisses and the amount of movement was recorded. Hisses were counted by reviewing the audio track of the video recording. Movement was scored by counting cell crossings on the video recording, defined as when four out of six feet moved to a new cell on the paper grid underneath the runway. Measurements were independently recorded after the experiment was complete by the experimenter and one observer.

Several modifications of methods (compared to Experiment 1) were made to facilitate hissing and reduce the likelihood of habituation to the US. These methods were selected based on *Thompson & Spencer*'s *(1966)* principles of habituation and

*Groves & Thompson*'s *(1970)* dual process theory. First, the acclimation period was reduced to 1 h. As collection may be a stressful experience, subjects were deliberately provided less time to recover, as agitated state may lead to sensitization instead of habituation. Second, the US was more intense. Habituation is more likely to occur to weak stimuli, so the gentle lift was used as a more ecologically valid mock predation attempt that was safe for the animals, but more intense than an abdomen touch. Finally, the intertrial interval was longer, as longer intertrial intervals delay the rate of habituation. Additionally, as more hissing was observed in the heated condition in Experiment 1, all subjects were heated in Experiment 2.

A 5-min preference assessment was conducted 30 min after the final conditioning trial. Drops of orange extract and peppermint extract were presented on opposite sides of the runway, with one acting as the CS and the other as a novel odor. The preference assessment was used to determine if an aversion developed to the CS odor outside of the initial conditioning method, as has been shown with bees (*Carcaud et al., 2009*). The duration subjects spent in each cell of the runway, with the two extreme ends containing a drop of odor extract, was assessed independently from the video recording by the experimenter and one observer. A subject was considered to be in a cell if at least four legs were in that cell on the paper grid underneath the runway. Duration measures were recorded to the nearest second. After completing the preference assessment, subjects were placed in a retirement bin, and the presence of hissing during recollection was recorded.

## Analysis

All regression analyses were conducted using the StatsModels package included in the Anaconda distribution of Python. Generalized estimating equations (*Hardin & Hilbe, 2003*) repeated measures regressions with exchangeable covariance structures were used to analyze behavior, as this approach provides robust, population-averaged estimates that account for within-subject correlation. In both experiments, a Poisson log link was used to analyze hiss rate, calculated by dividing the number of hisses by the measurement interval in seconds. In Experiment 1, movement was treated as a binary variable, and thus a logistic link was used to analyze the percent of individuals moving, expressed in the regression as probability of movement. In Experiment 2, movement rate was calculated and analyzed similar to hiss rate.

For all regression analysis, model selection was driven primarily by theory, focusing on core variables suspected to affect behavior, with standard model selection techniques used to probe additional predictors and two-way interactions. In Experiment 1, trial, sex, and a measure of temperature (either heating condition or actual temperature) were included in all models. Two-way interactions were investigated but were removed if they were nonsignificant ($p > 0.05$). In Experiment 2, trial and sex were included in all models. The odor and duration of CS along with all two-way interactions were investigated as possible predictors but were removed if they were nonsignificant. Higher-order interactions were not investigated in either experiment as they lacked theoretical support, and to reduce chance of overfitting. This model selection technique was used to explore potential variables and interactions that may affect behavior, while remaining closely connected to

| Table 1 Experiment 1 percent of cockroaches hissing. | | | | | |
|---|---|---|---|---|---|
| Group | Collection | Trial 1 | Trial 14 | Recollection | Never |
| Unheated female | 40.00 | 30.00 | 05.00 | 15.00 | 30.00 |
| Unheated male | 50.00 | 40.00 | 10.00 | 35.00 | 25.00 |
| Heated female | 30.00 | 50.00 | 05.00 | 20.00 | 25.00 |
| Heated male | 55.00 | 70.00 | 15.00 | 55.00 | 05.00 |
| All unheated | 45.00 | 35.00 | 07.50 | 25.00 | 27.50 |
| All heated | 42.50 | 60.00 | 10.00 | 37.50 | 15.00 |
| All female | 35.00 | 40.00 | 05.00 | 17.50 | 27.50 |
| All male | 52.50 | 55.00 | 12.50 | 45.00 | 15.00 |
| All cockroaches | 43.75 | 47.50 | 08.75 | 31.25 | 21.25 |

theory and transparently reporting even the absence of effect in core variables. The goal of this approach was to avoid overfitting issues that sometimes occur when stepwise selection techniques or an overemphasis on information criteria are used in the absence of theoretical considerations (*Henderson & Denison, 1989*; *Sutherland et al., 2023*; *Whittingham et al., 2006*).

Pearson's *r* correlations and *t*-tests were conducted using the SciPy package. In Experiment 2, Pearson's r correlation was used to analyze the relationship between hissing and movement during the conditioning procedure, and dependent *t*-tests were used to compare differences between durations subjects spent near odors in the preference test. In a *post hoc* analysis, an independent *t*-test was used to test collinearity between sex and weight.

## RESULTS

### Experiment 1: habituation

A simple overview of the percent of cockroaches hissing before, during, and after the experiment is summarized in Table 1. The data reveal that unheated subjects were less likely to hiss compared to heated subjects, and females were less likely to hiss than males. A decrease in likeliness of hissing across trials suggests habituation, further supported by the tendency for hissing to recover during recollection, indicating that the decrease was not due to fatigue. Twenty-one percent of cockroaches never hissed at all.

A detailed view of the rate of hissing during each 10-s trial is presented in Fig. 1. A decrease in hiss rate across trials can be observed for all groups. The overall rate of hissing is highest for heated male subjects, and lowest for unheated females. A corresponding statistical analysis can be seen in the top portion of Table 2, which considers heat to be a categorical variable (*i.e.*, unheated or heated). This analysis generally confirms what can be seen in Fig. 1. Habituation is observed for all groups (trial estimate = −0.148, $p < 0.001$). While heat appeared to influence the rate of hissing, this effect was nonsignificant (estimate = 0.567, $p = 0.054$). The role of sex was more substantial, with males hissing more (estimate = 0.691, $p = 0.019$). Given that the categorical heat variable may be a poor predictor of hissing since the actual temperatures of the animals varied, a second analysis,

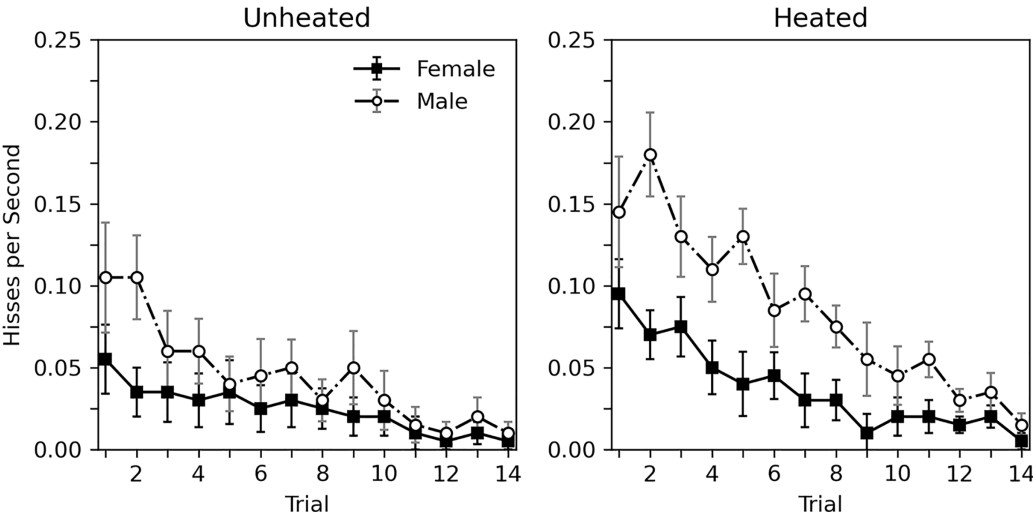

**Figure 1 Average rate of hissing in Experiment 1.** Average rate of disturbance hiss (hisses per second) across 10-s trials with brief abdomen stimulation for nonheated and heated subjects in Experiment 1. Error bars show standard error of the mean.

**Table 2 Experiment 1 hiss rate regression analysis.**

| Parameter | Estimate | Standard error | 95% confidence intervals | p-value |
|---|---|---|---|---|
| **Categorical heat** | | | | |
| Intercept | −2.806 | 0.310 | [−3.413 to −2.199] | <0.001 |
| Trial | −0.148 | 0.016 | [−0.180 to −0.117] | <0.001 |
| Heat | 0.567 | 0.294 | [−0.009 to 1.143] | 0.054 |
| Sex | 0.691 | 0.296 | [0.112 to 1.271] | 0.019 |
| **Continuous temperature** | | | | |
| Intercept | −3.910 | 0.677 | [−5.237 to −2.583] | <0.001 |
| Trial | −0.148 | 0.016 | [−0.180 to −0.117] | <0.001 |
| Temperature | 0.052 | 0.022 | [0.009 to 0.094] | 0.017 |
| Sex | 0.673 | 0.295 | [0.095 to 1.252] | 0.023 |

Notes:
For the categorical temperature analysis, unheated females are included in the intercept. The heat parameter can be interpreted as a change in rate of hissing when subjects are heated. For the continuous temperature analysis, females at 0 °C are included in the intercept. Temperature can be interpreted as the continuous effect of a 1 °C increase on rate of hissing. For both analyses, the sex parameter can be interpreted as a change in rate of hissing when subjects are male.

seen in the bottom portion of Table 2, replaced the categorical heat variable with a continuous temperature variable, derived from averaging the initial and final temperature of each subject. The overall trends remain; a significant decrease was observed across trials (estimate = −0.148, $p < 0.001$), hissing increases with temperature (estimate = 0.052, $p = 0.017$), and hissing is greater in males (estimate = 0.673, $p = 0.023$). The main difference between these analyses is that the continuous effect of temperature is significant. Interactions between variables are not reported in these analyses as initial explorations between trial, sex, and categorical heat or continuous temperature showed weak nonsignificant effects ($p$ values > 0.264).
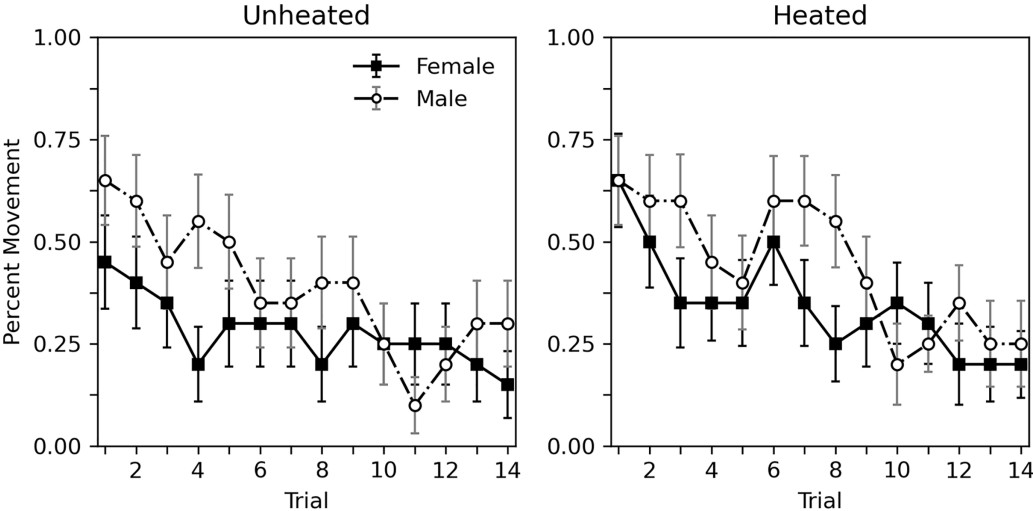

**Figure 2 Percent of subjects moving in Experiment 1.** Percent of subjects moving during 10-s trials with brief abdomen stimulation for nonheated and heated subjects in Experiment 1. Error bars show standard error of the mean.

**Table 3 Experiment 1 movement probability regression analysis.**

| Parameter | Estimate | Standard error | 95% confidence intervals | *p*-value |
|---|---|---|---|---|
| **Categorical heat** | | | | |
| Intercept | −0.164 | 0.262 | [−0.676 to 0.349] | 0.531 |
| Trial | −0.118 | 0.018 | [−0.154 to −0.083] | <0.001 |
| Heat | 0.326 | 0.275 | [−0.213 to 0.866] | 0.235 |
| Sex | 0.524 | 0.275 | [−0.015 to 1.063] | 0.057 |
| **Continuous temperature** | | | | |
| Intercept | −0.913 | 0.649 | [−2.184 to 0.359] | 0.160 |
| Trial | −0.119 | 0.018 | [−0.155 to −0.083] | <0.001 |
| Temperature | 0.034 | 0.022 | [−0.010 to 0.078] | 0.127 |
| Sex | 0.515 | 0.273 | [−0.020 to 1.049] | 0.059 |

Notes:
For the categorical temperature analysis, unheated females are included in the intercept. The heat parameter can be interpreted as a change in probability of movement when subjects are heated. For the continuous temperature analysis, females at 0 °C are included in the intercept. Temperature can be interpreted as the continuous effect of a 1 °C increase on probability of movement. For both analyses, the sex parameter can be interpreted as a change in probability of movement when subjects are male.

Figure 2 shows the percent of subjects moving during each trial. Similar to rate of hissing, some habituation of movement can be observed. Heating condition appeared to have little effect. Males moved more overall than females. Table 3 shows a corresponding analysis. As with Table 2, the top portion shows an analysis that considers heat to be a categorical variable, while the bottom portion uses a continuous temperature variable. When considering heat as a categorical variable, a decrease in probability of movement across trial can be observed (estimate = −0.118, $p < 0.001$). As can be seen Fig. 2, heat has no clear effect (estimate = 0.326, $p = 0.235$). Males may respond somewhat more than

Table 4 Experiment 2 percent of cockroaches hissing.

| Group | Collection | Trial 1 | Trial 8 | Recollection | Never |
|---|---|---|---|---|---|
| 3s CS female | 60.00 | 45.00 | 15.00 | 50.00 | 25.00 |
| 3s CS male | 60.00 | 75.00 | 35.00 | 70.00 | 10.00 |
| 6s CS female | 70.00 | 50.00 | 10.00 | 45.00 | 10.00 |
| 6s CS male | 70.00 | 90.00 | 30.00 | 70.00 | 00.00 |
| All 3s CS | 60.00 | 60.00 | 25.00 | 60.00 | 17.50 |
| All 6s CS | 70.00 | 70.00 | 20.00 | 57.50 | 05.00 |
| All female | 65.00 | 47.50 | 12.50 | 47.50 | 17.50 |
| All male | 65.00 | 82.50 | 32.50 | 70.00 | 05.00 |
| All cockroaches | 65.00 | 65.00 | 22.50 | 58.75 | 11.25 |

females, but this difference is nonsignificant (estimate = 0.524, $p$ = 0.057). These findings are reiterated when considering a continuous temperature variable. There is a significant decrease in probability of movement across trial (estimate = −0.119, $p$ < 0.001), heat did not affect movement (estimate = 0.034, $p$ = 0.127), and the differences between sexes was not significant (estimate = 0.515, $p$ = 0.059). As with hiss rate, interactions are not reported as initial explorations showed no significant interactions ($p$ values > 0.215).

## Experiment 2: classical conditioning

The percent of cockroaches hissing before, during, and after the experiment is summarized in Table 4. The percentage of subjects hissing at collection was notably higher than in Experiment 1. A general trend of habituation was observed, with hissing decreasing across the experiment and recovering during recollection. Eleven percent of cockroaches never hissed.

Figure 3 shows a detailed view of the rate of hissing during the pre-trial, CS, US, and post-trial intervals. Hissing never occurred during the 10-s pre-trial interval and only occurred once during the CS presentation. Hisses occurred regularly when the US was presented and continued to a lesser extent in the post-trial interval. The nearly complete lack of response to the CS, despite continual response to the US, suggests that no classical conditioning occurred. As in Experiment 1, males hissed more than females, and the tendency to respond to stimulation habituated across trials for both sexes.

Statistical analysis for the rate of hissing during the US and post-trial intervals is provided in Table 5. No analysis was conducted for the pre-trial and CS intervals due to the lack of response. Initial models included trial, sex, CS duration, and CS odor as parameters, but CS duration was removed from both analyses, and CS odor was removed from the post-trial hiss rate analysis due to high $p$ values. No interactions were significant in the initial models. The final analyses confirm what can be seen in Fig. 3; a significant decrease occurs across trials ($p$ values < 0.001) and males respond more than females ($p$ values < 0.001). Interestingly, CS odor did affect the rate of hissing during US presentations (estimate = −0.408, $p$ value = 0.014). Subjects exposed to peppermint as a CS odor actually hissed less during the overlapping US presentation.

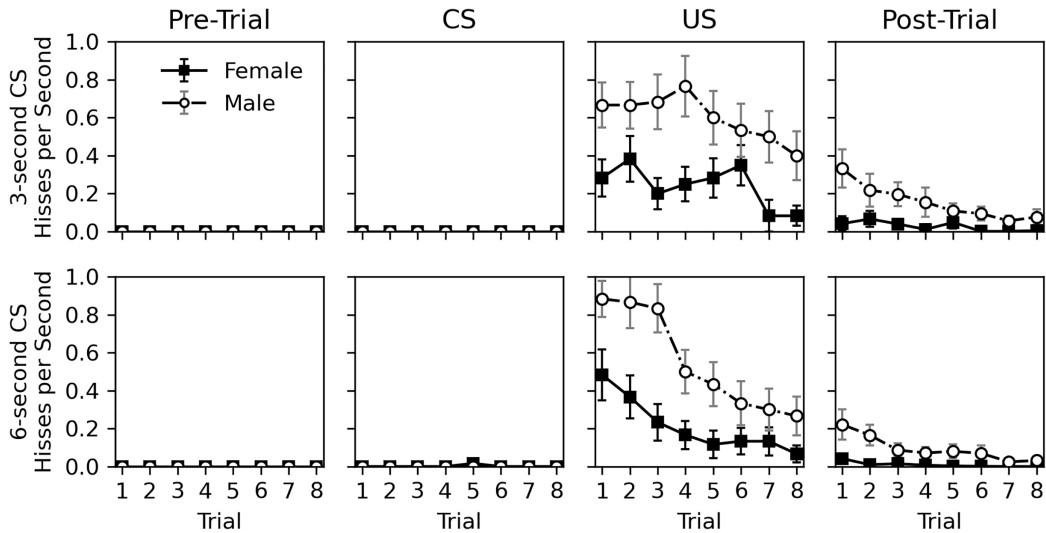

**Figure 3 Average rate of hissing in Experiment 2.** Average rate of disturbance hiss (hisses per second) across trials during the 30-s pre-trial intervals, 3- or 6-s olfactory CS intervals, 3-s tactile US intervals, and 30-s post-trial intervals in Experiment 2. Error bars show standard error of the mean.

**Table 5 Experiment 2 hiss rate regression analysis.**

| Parameter | Estimate | Standard error | 95% confidence intervals | *p*-value |
|---|---|---|---|---|
| **US hiss rate** | | | | |
| Intercept | −0.597 | 0.204 | [−0.996 to −0.198] | 0.003 |
| Trial | −0.143 | 0.025 | [−0.192 to −0.093] | <0.001 |
| Sex | 0.798 | 0.202 | [0.402 to 1.194] | <0.001 |
| CS odor | −0.408 | 0.166 | [−0.732 to −0.083] | 0.014 |
| **Post-trial hiss rate** | | | | |
| Intercept | −2.712 | 0.495 | [−3.682 to −1.742] | <0.001 |
| Trial | −0.273 | 0.042 | [−0.356 to −0.191] | <0.001 |
| Sex | 1.619 | 0.544 | [0.553 to 2.686] | 0.003 |

Notes:
For the US hiss rate analysis females receiving orange extract as a CS are included in the intercept. The CS Odor parameter can be interpreted as a change in rate of hissing when peppermint extract is used as a CS. For the post-trial hiss rate analysis, all females are included in the intercept. For both analyses, the sex parameter can be interpreted as a change in rate of hissing when subjects are male.

The rate of movement during the pre-trial, CS, US, and post-trial intervals is shown in Fig. 4. Movement was very rare during the pre-trial and CS intervals. When the US was presented, movement occurred but nearly ceased in the post-trial interval. Movement elicited by the US appeared to habituate across trials. Similar to Experiment 1, there appeared to be little difference in movement behavior between sexes. The near lack of movement during the CS interval, and its similarity to the pre-trial interval again suggests a lack of classical conditioning.

Table 6 shows statistical analysis for the rate of movement during the US and post-trial intervals. As with hiss rate, analysis was not conducted for pre-trial and CS intervals. Initial

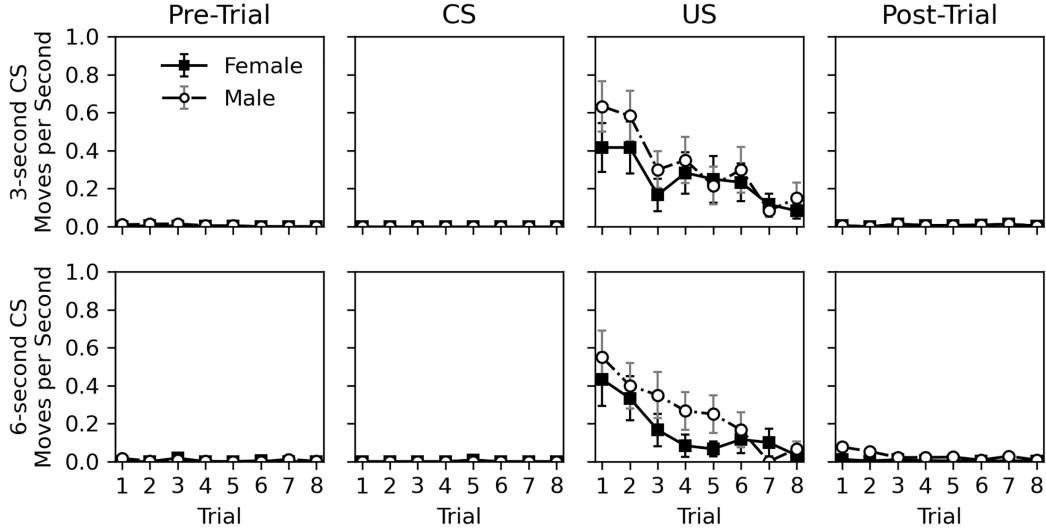

**Figure 4 Average rate of movement in Experiment 2.** Average rate of movement (cell-crossings per second) across trials during the 30-s pre-trial intervals, 3- or 6-s olfactory CS intervals, 3-s tactile US intervals, and 30-s post-trial intervals in Experiment 2. Error bars show standard error of the mean.

| Parameter | Estimate | Standard error | 95% confidence intervals | *p*-value |
|---|---|---|---|---|
| **Table 6 Experiment 2 movement rate regression analysis.** | | | | |
| **US movement rate** | | | | |
| Intercept | −0.617 | 0.195 | [−1.000 to −0.234] | 0.002 |
| Trial | −0.252 | 0.034 | [−0.320 to −0.184] | <0.001 |
| Sex | 0.367 | 0.203 | [−0.032 to 0.766] | 0.071 |
| **Post-trial movement rate** | | | | |
| Intercept | −4.907 | 0.402 | [−5.696 to −4.119] | <0.001 |
| Trial | −0.038 | 0.080 | [−0.195 to 0.119] | 0.633 |
| Sex | 2.596 | 0.474 | [1.668 to 3.524] | <0.001 |
| Trial * Sex | −0.249 | 0.100 | [−0.445 to −0.054] | 0.012 |

**Note:**
For both analyses, females are included in the intercept. The sex parameter can be interpreted as a change in rate of movement when subjects are male.

models explored CS duration, CS odor, and interactions between variables, but these were removed from final models due to lack of significance, except for a trial and sex interaction for the post-trial movement rate analysis. The analyses confirm that movement decreases across trials during the US interval (estimate = −0.617, $p = 0.002$). This is not true for the post-trial interval (estimate = −0.038, $p = 0.633$), likely due to such a low overall rate of response. Interestingly, the effect of sex was borderline. While males appeared to respond more during the US presentation, this effect was not significant (estimate = 0.367, $p = 0.071$). During the post-trial interval, males did respond more (estimate = 2.596, $p < 0.001$), and a significant interaction between trial and sex showed that this response decreased across trial for males (estimate = −0.249, $p = 0.012$). This interaction is likely

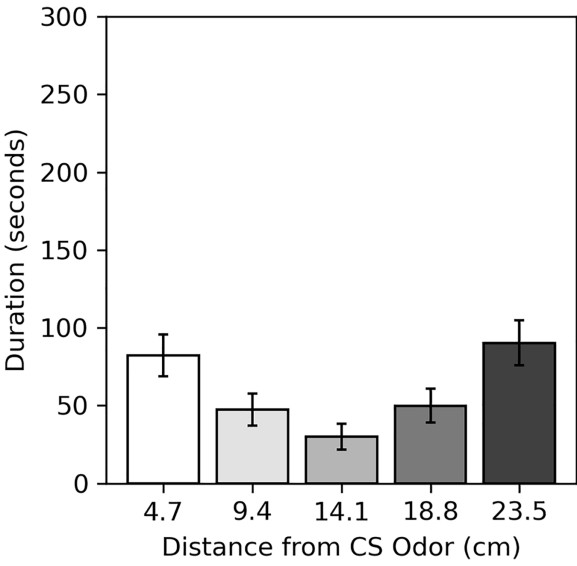

**Figure 5 Average duration spent near odors in the preference assessment of Experiment 2.** Average duration subjects spent within each distance from the CS odor in the preference assessment of Experiment 2. Distances are plotted in terms of distance from the previous CS odor, but they can be reversed to show the duration subjects spent near the novel odor. Error bars show standard error of the mean.

caused by the slightly higher initial rate of response for males in the post-trial interval, and the near zero level of response for females. Given the mixed significance and the very low rate of movement in the post-trial interval, the effects of sex should be interpreted with caution.

Pearson's $r$ correlation was used to explore the relationship between the total number of hisses and total amount of movement emitted during conditioning trials for each subject. The analysis revealed a strong significant correlation ($r = 0.554$, $p < 0.001$), indicating that individuals that hissed more frequently also tended to exhibit higher levels of movement. A similar analysis was not possible for Experiment 1 due to the binary nature of the movement data.

Results of the preference assessment conducted after the final conditioning trial are shown in Fig. 5. Generally, cockroaches were inactive during this assessment, as they were during pre-trial intervals. Many never moved at all (52.5% male, 60.0% female). Subjects spent the most time near the two extreme ends of the runway, regardless of odor. Dependent $t$-tests confirmed there was no difference in the mean duration subjects spent within 4.7 cm of the CS odor (82.24 s compared to the novel odor (90.26 s; $t(79) = -0.337$, $p = 0.734$), nor was there a difference in the mean duration subjects spent within 9.4 cm of the CS odor (129.75 s) compared to the novel odor (140.21 s; $t(79) = -0.345$, $p = 0.731$).

## Factors affecting the disturbance hiss
Both experiments found that the disturbance hiss was elicited by, then habituated to, tactile stimuli. Males produced more hisses than females, and some individuals never hissed at all. To further explore individual factors that may affect hissing, a *post hoc* analysis was

**Table 7 *Post hoc* hiss rate regression analysis.**

| Parameter | Estimate | Standard error | 95% confidence intervals | *p*-value |
|---|---|---|---|---|
| **Hiss rate 1** | | | | |
| Intercept | 1.835 | 2.401 | [−2.872 to 6.541] | 0.445 |
| Sex | 1.060 | 0.270 | [0.530 to 1.590] | <0.000 |
| Weight | −0.495 | 0.310 | [−1.103 to 0.113] | 0.110 |
| Temperature | −0.139 | 0.081 | [−0.299 to 0.020] | 0.087 |
| Weight * Temp | 0.021 | 0.011 | [0.000 to 0.042] | <0.050 |
| **Hiss rate 2** | | | | |
| Intercept | −1.375 | 0.750 | [−2.845 to 0.095] | 0.067 |
| Sex | 0.889 | 0.277 | [0.347 to 1.432] | 0.001 |
| Temperature | 0.000 | 0.024 | [−0.047 to 0.047] | 0.999 |

**Note:**

For both analyses, females are included in the intercept. The sex parameter can be interpreted as a change in rate of hissing when subjects are male. Temperature can be interpreted as the continuous effect of a 1 °C increase on rate of hissing.

conducted using data pooled across experiments. The analysis investigated the rate of hissing, defined as hisses per second during 10-s habituation trials in Experiment 1 or hisses per second during the combined 3-s US intervals and 30-s post-trial intervals in Experiment 2. Sex, weight, and temperature (defined as the average of initial and final temperatures for Experiment 1, and the initial apparatus temperature for Experiment 2) were included as predictors. The results are shown in the top portion of Table 7. The analysis indicates that, across experiments, males hiss significantly more than females (estimate = 1.060, $p < 0.001$). Weight does not appear to play a direct role (estimate = −0.495, $p = 0.110$), nor does temperature (estimate = 0.139, $p = 0.087$). However, the marginally significant interaction between weight and temperature (estimate = 0.021, $p = 0.049786$) suggests that while increases in weight may decrease rate of hissing, this effect is reduced as temperature increases.

In *G. portentosa*, females are generally heavier than males. Sex and weight therefore may be colinear. An independent samples *t*-test confirmed a significant difference in mean weight between sexes (females = 7.26 g, males = 5.95 g, $t(158) = 3.364$, $p < 0.001$). The analysis of hiss rate was therefore conducted again, this time excluding weight from the initial model. The results can be seen in the bottom portion of Table 7. After removing a nonsignificant interaction between sex and temperature, only the main effects of sex (estimate = 0.889, $p = 0.001$) and temperature remained (estimate = 0.000, $p = 0.999$). A similar analysis replacing sex with weight did not result in a significant effect of either weight or temperature (*p* values > 0.723). Together, these analyses suggest that sex is the most impactful individual factor on the disturbance hiss, with weight and temperature playing at most a secondary role. While the results of these exploratory analyses may be helpful in informing future work on the disturbance hiss, they should be interpreted with caution given the high collinearity between sex and weight and the fact that these data were not originally collected to address this specific question.

## DISCUSSION

### Habituation

Both experiments demonstrate habituation of the disturbance hiss and movement in response to aversive stimuli. While important to document, as this is only the third demonstration of learning in this species (and tribe), it is not an especially surprising finding given the highly conserved nature of habituation. However, differences between these experiments and those of *Davis & Heslop (2004)* provide valuable insights. First, in the present experiments, habituation was observed over the course of a single day, with intertrial intervals of 6 or 30 min. In *Davis & Heslop (2004)*, a 2-min continuous stimulus was presented once or twice daily, 6 days a week. Together this suggests that *G. portentosa* is capable of both short-term and long-term habituation, which may be distinct processes (*Rankin et al., 2009*). Second, *Davis & Heslop (2004)* reported that 40% of subjects never hissed, much larger than the 11% to 21% observed in the present experiments. There are a number of possibilities for this discrepancy, including sex and methodological variations. However, these findings converge on an interesting point; not all hissing cockroaches emit the disturbance hiss. Further investigations may consider if this is an individual trait or context dependent. In Experiment 2, the total amount of hissing was strongly correlated with the amount of movement, and across experiments, it appeared that males hissed more, suggesting investigations of individual factors may be a promising direction.

Future research on habituation should systematically explore the principles of habituation and sensitization outlined by *Thompson & Spencer (1966)* and *Groves & Thompson (1970)*. The fact that a higher rate of hissing was observed in Experiment 2 is a promising indication of the importance of factors such as intertrial-interval, stimulus intensity, and agitated state. Other species relevant factors may also be investigated. For instance, *Varnon & Adams (2021)* found the presence of food in habituation procedures momentarily inhibited startle responses in *Eublaberus posticus* cockroaches. Future work could also expand studies of habituation beyond the disturbance hiss and startle responses. Notably, habituation may be a substantial factor in reinforcer satiation (*Murphy et al., 2003*) and exploration (*Poucet, Durup & Thinus-Blanc, 1988*; *Wong et al., 2010*). Given the territorial behavior of male *G. portentosa*, which includes the agonistic hiss, research on habituation of conspecific aggression would also be a promising and biologically relevant topic. In some species, habituation has been suggested to be a mechanism for reduced aggression for known rivals (*Bee & Gerhardt, 2001*; *Petrinovich & Peeke, 1973*).

### Classical conditioning

The lack of classical conditioning in Experiment 2 was unexpected given the numerous studies on classical conditioning in cockroaches, other insects, and animals in general. There are several possibilities to consider. First, it is possible that classical conditioning cannot occur during habituation to the US. This appears unlikely. While there are no studies that specifically investigate classical conditioning during habituation, there is also no theoretical reason to exclude this possibility. For example, the Rescorla-Wagner model of classical conditioning (*Rescorla & Wagner, 1972*), actually predicts that learning will

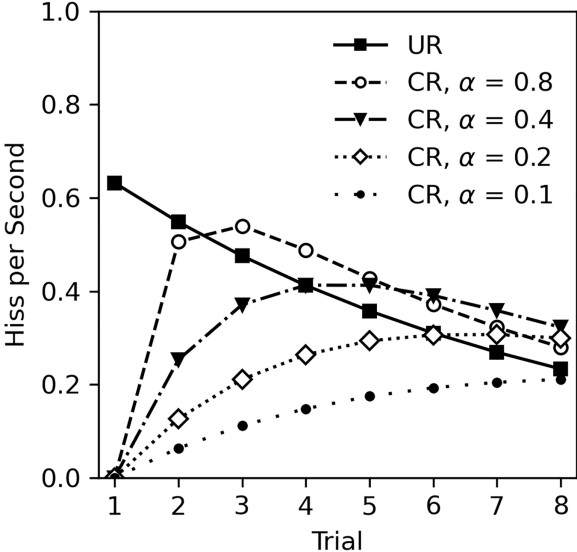

**Figure 6 Rescorla-Wagner model predictions for conditioned hiss rate in Experiment 2.** Rescorla-Wagner model predictions for rate of disturbance hiss (hisses per second) at several CS saliences ($\alpha$). The UR line was derived from analysis in Table 5, excluding sex as a factor.

occur during US habituation. The Rescorla-Wagner model is a quantitative model of associative learning that is extremely influential across behavioral sciences (*Siegel & Allan, 1996*; *Soto et al., 2023*). The core premise is that learning occurs when there is a mismatch between what is expected given the presence of a CS, and what actually occurs. In this experiment, the CS odor is presumably neutral and thus the subject does not expect tactile stimulation after detecting the odor. Touching the cockroach following odor presentation constitutes an unexpected change that will drive learning. Figure 6 illustrates Rescorla-Wagner model predictions based on the unconditioned response (UR) observed in Experiment 2. The UR line was derived from the analysis in Table 5, excluding sex and CS odor as factors, and functions as the US value in calculations. Conditioned responses are plotted at several levels of CS salience ($\alpha$) with a consistent learning rate ($\beta = 1$), assuming a one-to-one correspondence between US expectancy and hiss rate. Note that even with a small salience of 0.1, learning still occurs.

A second possibility is that the olfactory CS used were not sufficiently detectable. This is unlikely for several reasons. First, although conditioned responses were not observed, the odors used did affect the rate of hissing when the cockroaches were exposed to the tactile US. Subjects exposed to peppermint extract hissed less than those exposed the orange extract. Second, despite limited research on olfactory ability of *G. portentosa* and closely related species, evidence indicates that the antennae of *G. portentosa* respond to range of odors including 2-heptanone, 12-hexadecadienal, linalool (a component of orange extract), and lemon oil (*Szyszka et al., 2014*). Additionally, there is evidence that *G. portentosa* uses olfactory cues in food and mate selection (*dos Santos Benvenuto et al., 2024*; *Leibensperger, Traniello & Fraser, 1985*). Finally, cockroaches are generally

considered to have excellent olfactory abilities (*Persoons & Ritter, 1979*; *Schal, Fraser & Bell, 1982*). In classical conditioning research, conditioned responses to olfactory ability have been demonstrated primarily in the distantly related *Periplaneta* from the family Blattidae (*Arican et al., 2020*; *Hosono, Matsumoto & Mizunami, 2006*; *Matsumoto et al., 2013*; *Sato, Matsumoto & Mizunami, 2023*; *Watanabe et al., 2003*; *Watanabe & Mizunami, 2007*), but also in other members of the family Blaberidae that includes *Gromphadorina*, such as *Eublaberus* (*Varnon, Barrera & Wilkes, 2022*) and *Rhyparobia* (*Garren, Sexauer & Page, 2013*).

Another consideration is that perhaps these specific methods do not produce a conditioned response. This possibility is more likely and warrants future research. Experiment 2 demonstrated that a 3 or 6-s olfactory CS followed by an overlapping 3-s US then a 30-min intertrial interval was not sufficient to develop a conditioned response. This general method is common across olfactory conditioning research in cockroaches (*e.g.*, *Varnon, Barrera & Wilkes, 2022*; *Watanabe et al., 2003*) and other insects (*e.g.*, *Abramson et al., 2015*; *Busto, Cervantes-Sandoval & Davis, 2010*; *Desmedt et al., 2017*; *Frost, Shutler & Hillier, 2012*). It is possible, however, that a variation of these stimulus timings or other methods may prove successful. For example, restrained conditioning procedures work well for honey bees (*Frost, Shutler & Hillier, 2012*; *Varnon et al., 2018*), but are challenging for bumble bees and stingless bees, necessitating unrestrained modifications (*Muth et al., 2017*; *Amaya-Márquez et al., 2019*).

The final, and most interesting possibility is that this particular CS-US association may not be effective at developing hissing or movement as a CR. Such exceptions to the general rules of learning are not uncommon. A number of species-specific constraints on learning have been documented for both classical and operant conditioning. For example, *Garcia & Koelling (1966)* demonstrated that rats can learn to associate illness with taste or smell, but not visual stimuli. Guinea pigs, however, learn to associate both color and taste with illness (*Braveman, 1975*). Honey bees and vampire bats also have challenges learning to associate toxic substances with odor and taste (*Ratcliffe, Fenton & Galef, 2003*; *Varnon et al., 2018*). In the operant realm, *Bitterman (1965)* described a number of differences in how animals respond in serial reversal and probability matching experiments, and *Craig & Abramson (2015)* discussed how different species produce qualitatively distinct response patterns in fixed interval procedures.

Given the many findings of species-specific constraints on learning, the CS, US and responses need to be considered carefully. The CS odors, orange and peppermint, are derived from plants and may resemble food instead of being truly neutral. Unfortunately, little is known about the natural feeding habits of *G. portentosa* (*Schal, Gautier & Bell, 1984*). Presumably, they consume fallen fruits and other plant materials they find among the leaf litter. Laboratory diet studies show they will eat a variety of items including apples (*Carrel & Tanner, 2002*), carrots (*Mishra et al., 2011*), dog food (*Carrel & Tanner, 2002*; *Mishra et al., 2011*), and wheat bran (*dos Santos Benvenuto et al., 2024*). They may also be attracted to some plants, such as *Moringa*, but consume little of them (*dos Santos Benvenuto et al., 2024*). If the orange and peppermint odors used in this experiment

function as food related odors, then it may be challenging to associate these odors with an aversive tactile stimulus. Future research would greatly benefit from detailed olfactory studies to establish not only the ability of these animals to detect specific odors, but also to establish their neutrality or potential attractive or repellant effects prior to olfactory conditioning procedures.

The relationship between the US and potential responses is also important. In aversive conditioning procedures, species-specific defensive reactions are known to inhibit learning certain responses (*Bolles, 1970*; *Crawford & Masterson, 1982*; *Smith, Gustavson & Gregor, 1972*; *Smith & Keller, 1970*). It is possible, that for hissing cockroaches, hissing and fleeing are species-appropriate responses to a predation attempt, but not to stimuli that predict predation. Instead, tonic immobility may be a more adaptive response when potential threats are anticipated but not yet present (*Gallup, 1974*, *1977*). In other words, it may not be beneficial for a cockroach to hiss or flee if it has not yet been located by a predator. Detecting conditioned immobility in hissing cockroaches may be challenging, however, given their inactive nature. Note that subjects in Experiment 2 almost never moved during both pre-trial and CS intervals. Interestingly, despite a substantial body of work on olfactory aversive conditioning in other insects, there is limited corresponding research in cockroaches. The closest approximation is the use of salt solutions in differential taste conditioning procedures (*Varnon, Barrera & Wilkes, 2022*; *Watanabe et al., 2003*). Most procedures that focus on aversive learning follow the headless cockroach leg position method of *Horridge (1962)*, with a few studies using intact cockroaches (*Brown & Stroup, 1988*). Conditioned stimuli are not used in this work. Only a few studies explore the association of shock with other cues such as light or location (*Longo, 1964*; *Lovell & Eisenstein, 1973*; *Szymanski, 1912*). Clearly, more research in aversive conditioning is needed across cockroach species.

## CONCLUSIONS

These experiments aim to stimulate additional research on behavior and learning in hissing cockroaches. The work fills some important gaps, but also highlights the need for additional studies with *G. portentosa*, the disturbance hiss, and cockroach aversive conditioning. Given the lack of conditioning work with this species, it is crucial to report even null results, as describing differences and even a lack of behavioral abilities is an important component of research (*Avarguès-Weber & Giurfa, 2013*) and necessary to avoid publication bias and file-drawer effects often found in science (*Rosenthal, 1979*). Although research has repeatedly shown that learning is impacted by the ecological and evolutionary history of a species, even comparative research often focuses only on a small number of model organisms, leading to the neglect of entire orders and classes of animals (*Beach, 1950*; *Varnon, Lang & Abramson, 2018*; *Varnon & Moore, 2024*; *Vonk, 2021*). Continual work with *G. portentosa* offers an excellent opportunity to correct this deficit. Additionally, such research supports not only a scientific understanding of behavior, but also provides information crucial to furthering interest in cockroaches and other invertebrates as models to teach the next generation of scientists (*Abramson, 1986*; *Dixon et al., 2016*; *Matthews, Flage & Matthews, 1997*; *Proctor & Jones, 2021*).

## ACKNOWLEDGEMENTS

I would like to acknowledge Melissa Dandy (Oklahoma State University) and Georgina Fitzmaurice (South Carolina Governor's School for Science and Mathematics) for their contributions to experimental design and data collection.

### Funding

This project was supported by grant P20GM103499 (SC INBRE) from the National Institute of General Medical Sciences, National Institutes of Health. No additional external funding was received for this project. The funders had no role in study design, data collection and analysis, decision to publish, or preparation of the manuscript.

### Grant Disclosures

The following grant information was disclosed by the authors:
National Institute of General Medical Sciences, National Institutes of Health: P20GM103499 (SC INBRE).

### Competing Interests

The author declares that they have no competing interests.

### Author Contributions

- Christopher A. Varnon conceived and designed the experiments, performed the experiments, analyzed the data, prepared figures and/or tables, authored or reviewed drafts of the article, and approved the final draft.

### Data Availability

The raw data are available in the Supplemental Files.

The video demonstrating the basic habituation and classical conditioning procedures is available at Youtube: https://www.youtube.com/watch?v=u7-2TMZx1Mo. This video was created as a companion to the methods section of the article, after the experiment was complete. It was filmed in a quiet room with full lighting to provide the clearest view of the procedure possible. For the purpose of demonstration, some aspects of the procedure, such as the inter-trial intervals, were omitted.

### Supplemental Information

Supplemental information for this article can be found online at http://dx.doi.org/10.7717/peerj.19805#supplemental-information.

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
