# Peer review of "Habituation but not classical conditioning of the disturbance hiss of the hissing cockroach (Gromphadorhina portentosa)"

_PeerJ, doi:10.7717/peerj.19805_

## Round 0.1 · original submission · Major Revisions

All reviewers appreciate the study but require additional information (methods not clear) or extra explanations. Some of the points are highlighted to be critical to address before acceptance, esp reviewer 3 raises some important aspects. Figures and overall methods need much more information/ clarity to be able to reproduce the study. Improve figures which are supposed to be with the caption self-contained. It further seems the study requires a more in-depth discussion.

·

Basic reporting

1. Writing is often ambiguous. See comments on pdf for examples.
2. The author has done a nice job of using supporting literature. However, that literature can be used more effectively. For example, use the published literature to justify the experimental design. Also, explain the relevance of cited literature instead of giving a long list of cited papers. Explain how each paper is relevant to your work.
3. The figures need more informative figure captions. A good rule to follow is the figure caption and figure taken together should be informative independent of the text.
4. Each figure needs error bars.
5. The y-axis on each figure needs a more informative label. For example, "hiss rate" is ambiguous. Explain its meaning in the figure caption.
6. Use chronological order when listing multiple citations at one time.
7. I would like justification for the exploration of heat and sex on cockroach behavior. Temperature in particular was not discussed in the introduction. There are logical reasons to expect temperature to play a role in cockroach behavior, but you need to explain them.

Experimental design

I found the experimental design to be well defined and appropriate for the research question. The author did a great job identifying a knowledge gap.

1. As commented on the pdf, I would like the papers used to develop the methods cited in the methods. For example, I wondered why the specific odors were used but had to wait until the discussion to find the justification for them.

2. Be consistent on your terminology for the response variables. Be clear what each one is measuring. For example, I argue you did not calculate "probability of movement" but rather "percentage of individuals that moved." The latter more accurately describes what you measured and is unambiguous.

Validity of the findings

The author did a good job of discussing the findings in context of the published literature. I liked the post-hoc analysis of the Rescorla-Wager model, as it demonstrates the author is strongly considering how his results fit into existing models of conditioning. However, the author should provide more explanation of the Rescorla-Wager model. The model is introduced with little explanation.

See additional comments on pdf.

Reviewer 2 ·

Basic reporting

The English language used is clear and concise. The paper's organization is also appropriate, and the literature is reviewed with adequate care. However, regarding the details of the experiments, the paper does not provide adequate information to clarify methods and procedures. This issue is particularly concerning as there is no clear exposition of the experimental setup and how the measurements were obtained. The paper greatly benefits from including specific figures and explanations, showing how the experiments were performed, and providing details of how the measurements were obtained. Doing so will help further investigations in the future.
Also, I am not sure whether all tables should be included in the main text. Perhaps, some of them could be moved to the supplementary materials.

Experimental design

The paper presents the results of two experiments, investigating changes in the disturbance hiss of the hissing cockroach (Gromphadorhina portentosa). Two specific experiments were conducted to study the habituation behaviour in G. portentosa, where the first experiment investigated habituation to repeated tactical stimulus delivered to the cerci; the second experiment explored classical conditioning by presenting an olfactory stimulus. The hissing and the movements were measured during these experiments. While research questions are clear, and the results provided are encouraging, lack of sufficient information about the methods and experiments make it difficult for me come to a close evaluation of the work. In this sense, I am not sure if the information provided is sufficient for replicating this study.

Validity of the findings

All data recorded during the experiment is provided in the supplementary files. However, I would encourage the author to include videos in the supplementary files to provide further proof for supporting the results presented in the paper (if available) and demonstrating the hissing behavior in a more detailed manner.

Additional comments

The index presented in Figure 2 requires further elaboration. Please include an extra explanation in the paper and describe how the probability of movement is calculated.

Figure 5: please clearly demonstrate how the distance from the CS Odor is measured, given that at different times, the distance could be different. Please elaborate on your convention in measuring the distance, and at which instant the measurements were obtained.

Line 85: “in terms learning,” shouldn’t it be “in terms of learning”?

·

Basic reporting

The ms. "Habituation but not classical conditioning...hissing cockroach (G.p.)" presents a study of learning in a hissing cockroach species that is commonly used in educational settings such as undergraduate behavior classes. Its social communication through agonistic hissing as well as olfactory cues has been fairly extensively studied in the lab setting, The current study was motivated by a lack of experiments on its learning abilities, especially its response to classical conditioning of its presumed anti-predator response, the distinctive "disturbance hisses". It presents the results of two laboratory experiments, each with a laudably large number of subjects (80/experiment-broken evenly into groups by sex or temperature), one demonstrating habituation of the hissing response across repeated trials during a short time and the other designed to show whether the disturbance hissing or avoidance (movement away) could be conditioned to occur in response to an otherwise non-meaningful odor.

Overall, the study is clearly written, easy to follow, and structurally in accordance with published standards.
The Introduction summarizes many of the topics on which lab studies of this or other cockroach species have been done, including those on learning and conditioning in other species and motivates the current report in terms of there being only two studies of habituation in G. portentosa and none of their response to conditioning.
The Methods and Results are generally clearly written, although a number of improvements should be made or dangling questions answered (see more below).
In general, relevant literature is covered, although note one small but nevertheless useful study that went uncited (Shotton 2014 Testing the disturbance hiss of the Madagascar hissing Cockroach (Gromphadorhina portentosa) as an anti-predatory response. Bioscience Horizons 7 doi 10.1093/biohorizons/hzu010). I will return to this below.
The figures are easy to read and useful on the face of it. However, they will need much more extensive, carefully documented legends to explain each one. Also, the naming of the y-axes in particular should include units and ascribe as much as possible to the variable names used in the text. Where the metrics have not been clearly described, the legend should spell them out. In the case of bar graph Figure 5, I believe that mean durations are displayed and thus error bars are required. Overall, as they are labeled and with their current legends, the figures do not provide sufficient stand-alone information. Where relevant, specific needed clarifications are mentioned below.
Supplementary Data set: The data sets appear complete and easily open as csv files, a good format for posterity. However, see the Validity of Findings section for several needed changes and questions arising from perusing the complete data set.

Thus, while the study is presented clearly in general, there are numerous points for improvement, ways in which this could be a better motivated, better documented and presented, and more usefully interpreted study. As I will point out below, this starts with a more thoughtful and impactful motivating introduction, something more than "it hasn't been done for this species".

Experimental design

My comments on the experiment will follow the sections of the paper.

Introduction: The clear focus of this study is the ability to learn in an insect, including demonstrating general habituation of this behavior (not surprising, as the author notes) and conditioned pairing of the behavior with a novel stimulus. The introduction is extremely narrow and motivates the study primarily in terms of the lack of such a previous study on this particular species, despite its being a "well-known" laboratory insect used in many science education contexts. The main argument is that topics related to learning have been extensively studied in "other cockroach species", but not G.portentosa. As a biologist and behavioral ecologist, I found this inadequate. We know a good deal about cockroach phylogeny and about the ecology of some species, but in fact, little has been published on the natural history of G. portentosa. Notably Staddon among many others have pointed out the important connections between learning abilities and species-typical ecological and social contexts. Comparing ecological context and learning patterns can work forward and backward to give us insights on adaptive learning or possible behavioral patterns or challenges in the wild. Phylogeny might also be related to cognitive abilities as adaptive features of particular lineages. Highlighting previous results on more closely related species or species more similar in morphology or social behavior would provide a way to set up some expectations. Obviously habituation is found reliably across many animals--but different behaviors easily habituate or not, depending on their environmental context. In any case, the introduction could be broader, more biological (ecological, evolutionary) and thus link better with the last explanation in the Discussion that the author finally comes to with respect to explaining an apparent lack of conditioning. This is clearly the most interesting and also most justifiable type of explanation, while tweaking of timings and intervals could go on forever.

With respect to the introduction and setting up the possible value in this study, the author has missed an interesting, if limited study by Shotton (see full reference above) It suggests that the disturbance hiss might be more than just anti-predator and, in fact, notes and discusses the same immobility response in the face of a predator or predator-stimulus as found in this study. It would have been useful to elaborate on what we might not know about anti-predator behavior and the exact contexts in which the disturbance hiss is adaptive, and then build on that to set up in the Introduction the alternative predictions for how a classical conditioning experiment might fail to measure a conditioned response that was there (immobility as a response). Furthermore, a small data set included with Shotton (2014) also showed the sex difference in hissing reported here.
This points up another issue: heat or body temperature and sex were included in this study as variables, but the justification or the hypothesized differences they should make were neither explained or put in terms of predictions or expectations. They are just mentioned in the Intro.

With respect to classical conditioning, lines 98-102 seem to be building toward some intrinsic interest in G.p.'s abilities, but the sentence appears incomplete, and by the end of the Intro, we are back to investigating it because no one has yet. There is no exploration of the possible ecological context for conditioning this particular behavior. What would it mean to demonstrate aversive learning or not? The intro ends with no real predictions or hypotheses or background theories as context for learning abilities expected of G.p.

Methods
The methods seem complete on first reading but in fact, a number of details that are likely to be important to this behavior should be added.
1. For the basic background on the colonies and individuals tested, what was the length of time that the colonies had been maintained as such and how crowded were they (if social context could affect readiness to hiss and to habituate)? If 80 were taken out of a single colony in a box of the size described, they must have been fairly crowded. What was the handling and transfer history of the colonies--c.f. Davis and Henslop 2004?
2. Were the colonies actively breeding and were the individuals actually adults, i.e., past their last instar? Again experience with social disturbance, ability to avoid others and amount of prior handling provides a background context for hissing easily or less easily elicited, especially if Shotton 2014 and prior studies are right and the disturbance hiss is not simply directed at predators.
3. When handled for this study, was only one experimenter involved using a standardized procedure (holding by thorax, etc)? The holding method is described for Exp 2 but not for Exp 1. Were hand lotions and perfumes controlled?
4. The Methods state that duration and count of hissing was not recorded for Exp 1. But line 150 states that the number of hisses for the trial was recorded. And in the second experiment, hisses were counted. How was "a hiss" defined? How were they counted--by ear at the time or from audio recordings later?
5. Temperature differences in Exp 1 are important. But the Supplementary data sets show a great deal of variation and change in temperatures during the experiment. How were the means in L 138 computed--from the initial or final temperature readings. (Standard deviation and range??) See Validity of Findings for a full set of questions on this variable.
6. Exp. 2's apparatus was open to odors from the room at all times? Clarify what the fume extractors were doing? Was the same apparatus used over and over and, if so, was it wiped down with alcohol between uses?
7. The temperature during Exp 2 was approximately that of the heated temperatures of Exp 1. Why?
8. L199. Clarify—‘response to recollection’ This is an actual measure. Was it simply y/n hissed? any counts or “vigor” or duration? WHO did the recollection…the person presenting the stimulus CS odor?
9. Statistical analysis. (Are there missing words in L202-203?) although stats are given here, one should still clarify the statistical test/approach when reporting the results in the text. Note that key summary variable "rate of hissing" is never defined. It appears to be the Proportion (fraction?) of Individuals that Hissed...not a rate at all! Likewise "probability of movement" first encountered in the results (L 223) is in fact the proportion of cockroaches moving, correct? Clarify the metrics used within and across subjects before we get to Results.

Results:
1. This reviewer does not find the descriptive overview of results useful, since it is not clear what comparisons stand up in statistical analysis or not. This goes for the result report of both experiments.

2. Exp 1. Effect of temperature: See Validity of Findings; the data sets suggest that the actual temperatures at which individual cockroaches were behaving and being tested were quite highly variable and the two temperature categories were not clearly divided, with individuals in one category overlapping in body temp with those in the other. This needs to be addressed given the significant but "marginal" effect reported. A cleaner difference could have greatly increased its import.

3. L 221-222 If interactions were explored, why not show the results? (l 222--the stat term is NON-significant, not unsignificant.)

4. Exp 2. The comparison of hissing with those in Exp 1 is interesting, especially given the apparent differences in average body size in Exp 2, which had many more smaller cockroaches. Body size (weight) was measured but never explored. Perhaps it should be, as it is one thing that generally separates females (larger) than males.
5. Exp 2. L 263-265. Why are the time reported for two measures of distance from the odor?
Discussion
1. If movement and hissing # were correlated, how did they vary with temperature? Although individuals' temperatures were recorded and varied in Exp 2, nothing was done with the data. But this seems a likely link between the two active responses.
2. In discussing the lack of conditioning and detectability of odors (l 317), this question should just not come up at this point. The odors should have been validated (they were detectable--e.g. by movement of antennae or other evidence) or justified as chemically reasonable. But I would agree that it was unlikely. Note that L 319-322 are confusing, since they seem to suggest that other classical conditioning studies with olfaction have been done of G.p. Clarify..other species--which??? close relatives of G. p.?
3. Again, because the ecological contexts and adaptive aspects of learning were not addressed in the Intro, there is less of an entrance to the more productive line of explanation given at the very end...that the cockroaches ARE responding, but with immobility. I would shorten the other post-hoc explanatory material. One might also end with suggestions for a broader study of hissing and immobility as behavioral responses in relation to (pseudo) predatory events--tying in the sex differences and thermal effects. A more evolutionarily (and physiologically) informed introduction would be a better set up for adaptive explanations for the results and impetus for tying learning to this species' ecological and social context.

Validity of the findings

First, as suggested above, changed emphasis and background in the Introduction will give more impetus to the study and increase the interest for future work in the Conclusions. That said, several aspects of the current study could be investigated more thoroughly with the data set at hand could clarify the effects of sex and temperature, and perhaps add body size.

Skimming the actual data sets raised several questions not dealt with or mentioned in the text. In the Exp. 2 data set, I note a female of only 2.95 gm. This is quite small for an adult female (although they do have quite a range) and this brings up the question (noted above) of whether indeed all tested cockroaches were adult and how this was known. (A reassurance is all that is needed; it is relatively easy to tell.) But it is strange that the weights in Exp 1 appear to be overall greater than in Exp 2. I also note that all weights in Exp. 1 were in rounded (whole) numbers while the Exp 2 weights are accurate to 2 places. Was there a strong difference in how they were weighed? Was there a reason for Exp 2 individuals to be more frequently <5 gm? I gather that size was never tested as a variable in hissing or moving behavior. Predation risk, mobility and cost of hissing loudly might all be tied to body size.

Finally, the starting and ending temperatures of heated and unheated cockroaches have a lot more variation than I expected from the Methods. Yes, different sized cockroaches will cool at different rates. But some unheated cockroaches in particular and a few heated ones apparently increased in temp during their trial periods in the test apparatus, markedly in some cases (nearly 5 deg F) or dropped almost 20 degrees! (e.g Cockroaches no 41, 42 in Exp 1.) The result was that some heated ones were colder than unheated ones by end of their habituation trials. Are the room temps changing or were there heating pad variations?? Any explorations of actual temps of cockroaches and their hissing or moving propensity?? Certainly, elevated temperatures affect the frequency of social activity and social hissing as well as metabolism.

Figures and Tables. Must have complete legends that clarify everything about the figure or table...the metrics, the message of the figure/table, where the statistics are reported if not there.


Supplementary Data set: These files need more explication to be used, even with the Methods accessible. Thus, the data provided with a final paper should be accompanied by a ReadMe file explaining all variables, to make them interpretable apart from the paper Methods section. It would also be helpful if the variable names (e.g. Initial Hiss or Final Hiss) were the same as those used in text or figures in the paper itself. Specify in the data set which variables are Y/N and thus either 1 or 0 (categorical rather than continuous variables). Variable names can be longer in these files for the sake of clarity. Anyone wanting to repeat or extend the analysis can apply their own short names. Complete clarity and documentation is the goal here.

Units in the supplementary data set should be specified within variable names. All units should be those in standard scientific use, i.e., in Celsius (temperature) rather than the Fahrenheit shown. The weights appear to be correctly metric (grams). As they are, the Supplement spreadsheets are not stand-alone data sets.

Additional comments

This study has more to offer in terms of results as well as in terms of ideas for further experiments if recast in more "adaptive" terms. More complete use of the individual variability in size and temperature may reveal patterns related the issue of immobility versus escape or avoidance behaviors. It would be a shame and a misrepresentation of the cost-benefit modulation of behavior that is likely contributing to individual variations, from responding or not to rapidity of habituation or hissing to one form of stimulation vs the other.

I am in entire agreement that "negative" results should be published, but in this case, what is negative just stems from not allowing for multiple outcomes. The results are not "negative", they are alternative. It can be a very useful step in understanding this species and comparing its response to those of species with known ecological situations.

---

## Round 0.2 · Minor Revisions

Most of the comments had been addressed, yet two issues must be addressed prior to getting acceptable

1) future experiments MUST verify odour stimuli first, to make sure these odours are not meaningful in some way or at least do not elicit different degrees of avoidance, attraction or attention.

2) The authors should provide a video or a schematic / series of images to 'visualise how the experiments are conducted'. The data should be made accessible together with supplementary data and files in a depository. Scientific data, and most data, as a rule of thumb, must be at least stored for 5 years; it is not acceptable./professional to lose this data - if this is the case you must provide a statement in your draft in a section 'data availability' and you should justify which data is missing. A final decision will also be made based on the completeness of raw data in a data repository.

·

Basic reporting

The author adequately addressed the comments of the original three reviewers.

Experimental design

The author adequately addressed the comments of the original three reviewers.

Validity of the findings

The author adequately addressed the comments of the original three reviewers.

Reviewer 2 ·

Basic reporting

The author took a large effort to improve the basic report of the paper, and I am satisfied with the revised version.

Experimental design

I have requested the author to provide some figures or videos to visualize how the experiment is performed. I believe this issue is reasonable and necessary, which facilitates replicating the results of the experiments. Unfortunately, the author has refused to do so, making some excuses about the loss of hard drive data, which does not make sense in a professional setting.
I acknowledge that some changes were made to clarify the experimental design within the text, but providing photos/videos from the experiment is standard and required.

Validity of the findings

I have no further comments as to the validity of findings.

·

Basic reporting

The rewrite of ms. "Habituation but not classical conditioning...hissing cockroach (G.p.)" has improved and problems highlighted in my original review have been corrected or clarified. The references have been expanded. Figures and legends have been redone as needed.
There are several things still not quite fixed. The ranges of weights for the two sexes still do not reflect the actual ranges shown in the raw data, as there are cockroaches below the lower bound for both males and females in the data set.
Also, line 412 documents a significant difference with sex, females being heavier, but the actual means are reversed, with males shown as heavier: [females = 7.26g, males = 8.95g, t(158) = 4.458, p < 0.001)].
(= one minor revision)

Experimental design

The introduction is somewhat broader and better justified, and tied well to the discussion. I would probably have tried to tie it to natural history more, but this is adequate and the discussion with respect to habituation and conditioning procedures and expectations is fine.
I was happy to see an exploration of weight and temperature as continuous variables.

Note that the two odors used in conditioning should have been tested prior to use. This cannot be changed, of course. See 3. below

Validity of the findings

In general, statistically sound, and one appreciates the care not to over-fish for patterns.
However, given the significant difference in responses to the two odors used as "neutral" conditioned stimuli, the author should really comment that future tests MUST test those stimuli first, to make sure the odors are not meaningful in some way or at least do not elicit different degrees of avoidance, attraction or just attention. Given that they are forest floor feeders on fallen fruits among other things, chemical extracts of potential food might be expected to be non-neutral. The problem of pairing vaguely food related odors (orange plus peppermint is extracted from a common family of plants and this species does eat green plants) with predator-stimuli is a reasonable point of discussion, and a reasonable hypothesis to investigate further.
I suggest strongly that a) the author admonish future experimenters to test for "neutrality" first and b) there be a mention of food associations given what this species eats--which is NOT dog food in Madagascar! (=one minor revision)

Additional comments

The revision seems complete and the resulting ms, with exception of these points, appears clear.

---

## Round 0.3 · accepted · Accept

The authors have addressed all comments of the reviewers.

Reviewer 2 ·

Basic reporting

No further comments on basic report.

Experimental design

The video is uploaded on YouTube, which might be lost over time. A link to a permanent repository is preferred (Zenodo).

Validity of the findings

No further comments.